# ZrO_2_ Nanoparticles and Poly(diallyldimethylammonium chloride)-Doped Graphene Oxide Aerogel-Coated Stainless-Steel Mesh for the Effective Adsorption of Organophosphorus Pesticides

**DOI:** 10.3390/foods10071616

**Published:** 2021-07-13

**Authors:** Xiudan Hou, Rong Ding, Shihai Yan, Haiyan Zhao, Qingli Yang, Wei Wu

**Affiliations:** 1College of Food Science and Engineering, Qingdao Agricultural University, Qingdao 266109, China; qdxdhou@qau.edu.cn (X.H.); dingr1025@163.com (R.D.); xinyuyuanyin@163.com (H.Z.); yql@qau.edu.cn (Q.Y.); 2College of Chemistry and Pharmaceutical Sciences, Qingdao Agricultural University, Qingdao 266109, China; shyan@qau.edu.cn

**Keywords:** graphene aerogel, ZrO_2_, solid-phase extraction, organophosphorus pesticides

## Abstract

A novel sorbent based on the ZrO_2_ nanoparticles and poly(diallyldimethylammonium chloride)-modified graphene oxide aerogel-grafted stainless steel mesh (ZrO_2_/PDDA-GOA-SSM) was used for the extraction and detection of organophosphorus pesticides (OPPs). Firstly, the PDDA and GO composite was grafted onto the surface of SSM and then freeze-dried to obtain the aerogel, which efficiently reduced the accumulation of graphene nanosheets. It integrated the advanced properties of GOA with a thin coating and the three-dimensional structural geometry of SSM. The modification of ZrO_2_ nanoparticles brought a selective adsorption for OPPs due to the combination of the phosphate group as a Lewis base and ZrO_2_ nanoparticles with the Lewis acid site. The ZrO_2_/PDDA-GOA-SSM was packed into the solid-phase extraction (SPE) cartridge to extract OPPs. According to the investigation of different factors, the extraction recovery was mainly affected by the hydrophilic-hydrophobic properties of analytes. Effective extraction and elution parameters such as sample volume, sample pH, rate of sample loading, eluent, and eluent volume, were also investigated and discussed. Under the optimal conditions, the linearity of phoxim and fenitrothion was in the range of 1.0–200 μg L^−1^, and the linearity of temephos was in the range of 2.5–200 μg L^−1^. The limits of detection were ranged from 0.2 to 1.0 μg L^−1^. This established method was successfully applied to detect OPPs in two vegetables. There was no OPP detected in real samples, and results showed that the matrix effects were in the range of 46.5%–90.1%. This indicates that the ZrO_2_/PDDA-GOA-SSM-SPE-HPLC method could effectively extract and detect OPPs in vegetables.

## 1. Introduction

Pesticide residue is an importantly concerning aspect of food safety. Nowadays, the population of food poisoning caused by pesticides occupies one-third of the total number of people of food poisoning, among which organophosphorus pesticides (OPPs) take the first place. OPPs, as a class of agricultural chemicals, are mainly used to prevent the invasion of insects. However, excess of OPPs remain in the surface and body of agricultural products, infiltrate into the environmental samples, and furthermore, enter into the human body, which can cause headache, dyspnea, and dysphoria symptoms [1,2]. The long-term consumption of these pesticides can lead to cancer and even death [3]. China and other countries have stipulated that the maximum residue limit of most OPPs should not be higher than 0.2 mg kg^−1^ or 0.5 mg kg^−1^ [4]. Therefore, it is important to detect and monitor OPPs in food in order to guarantee the food safety.

Due to the complexity of matrices in food samples, the diversity of ingredients, and trace residue level, it is exceedingly difficult to detect OPPs directly making use of an instrument. Generally, a suitable sample pretreatment method is chosen to separate and enrich target OPPs prior to using the instrument in order to improve the sensitivity and accuracy. Traditional extraction methods including liquid–liquid extraction and solid-phase extraction have some disadvantages such as being highly organic solvent consuming and time consuming, etc. In recent years, many new material-based extraction methods have been introduced to detect OPPs, e.g., dispersive liquid–liquid microextraction [5], hollow-fiber liquid-phase microextraction [6], dispersive micro-solid phase extraction [7], magnetic solid-phase extraction [1], ultrasound-assisted emulsification liquid-phase microextraction [8], fiber solid-phase microextraction [9], etc., which can shorten the extraction time and minimize the solvent consumption. Nowadays, the studied extraction materials mainly include metal organic frameworks [10,11], covalent organic frameworks [12], layered double hydroxides [13], molecularly imprinted polymers [14], graphitic carbon nitride [15,16], and ionic liquid [17,18,19].

The stainless-steel mesh-based sorptive extraction was introduced as a new extraction method [20]. A large amount of adsorption materials can be coated onto the surface of meshes to greatly increase the contact surface area, and the extraction equilibrium is relatively faster.

Graphene aerogel (GA) has attracted more and more attention in different fields since its emergence in 2013 towing to the super large specific surface area, excellent mechanical properties, high porosity, and low density [21,22,23]. As a desirable sorbent, porous and amphiphilic GA was designed and prepared for the adsorption and preconcentration of environmental pollutants [24,25,26,27]. Although the 3D frame structure of GA can provide many adsorption sites, it has the disadvantage of a single structure with the poor adsorption selectivity for OPPs. Therefore, in order to achieve high-efficiency enrichment of trace OPPs in complex food samples, it is necessary to prepare functionalized GA to improve the extraction performance. Furthermore, it can be realized through the framework structure, oxygen-containing groups, and the interaction of the carbon skeleton of graphene oxide aerogel (GOA). However, it remains to be solved that GA not only possesses good mechanical strength, but also offers excellent extraction performance for OPPs. Some polymers are capable of elevating the mechanical features of GA [28,29]. For example, Bai et al., fabricated a porous GOA using polyethyleneimine as the cross-linking agent to avoid the collapse of the structure, which was successfully used to extract thorium (IV) from three lanthanides (III) elements [30]. Poly(diallyldimethylammonium chloride) (PDDA) was regarded as a cationic polyelectrolyte due to the abundance of positively charged quaternary ammonium groups, which were widely used in many fields [31,32,33]. PDDA could be introduced to prevent the irreversible agglomerate of graphene nanosheets and also enhance the mechanical stability to avoid the collapse of the aerogel structure.

Zirconia (ZrO_2_), as an inorganic oxide, exhibited excellent properties of high specific surface area, chemical inertness, thermal stability, and lack of toxicity [34]. It has been reported that ZrO_2_ possesses a strong affinity for the phosphoric group, and the phosphate group as a Lewis base could strongly interact with the Lewis acid site on the surface of ZrO_2_ to form coordination bonds, which makes ZrO_2_ as a selective sorbent for the detection of OPPs [35]. ZrO_2_-based sorbents were used for the selective determination of OPPs in real samples [36,37,38]. Individual ZrO_2_ nanoparticles were easy to aggregate. In order to expose more adsorption sites, nanosheet-structured graphene as a desirable material could be utilized to improve the dispersity of ZrO_2_ nanoparticles.

This work prepared a ZrO_2_ nanoparticles and PDDA-modified GOA-grafted stainless-steel mesh (ZrO_2_/PDDA-GOA-SSM) as the sorbent for the extraction of OPPs. The material enhanced with PDDA exhibited higher stability. This sorbent device integrated the advantages of the sorptive extraction and the adsorption property of GOA. Various surface analysis techniques were used to characterize the morphology, structure, and composition of ZrO_2_/PDDA-GOA-SSM. A SPE column was packed with ZrO_2_/PDDA-GOA-SSM to extract OPPs. The applicability of the established ZrO_2_/PDDA-GOA-SSM-SPE-HPLC method was also evaluated.

## 2. Materials and Methods

### 2.1. Materials and Reagents

The stainless-steel mesh (SUS304, pore 74 μm, Φ 50 μm) was purchased from Leiko Metal Products Co. LTD (Changzhou, China). Graphite powder, phoxim (99%), temephos (99%), fenitrothion (99%), 3-aminopropyltriethoxysilane (APTES), PDDA (MW = 200,000–300,000), N-hydroxy succinimide (NHS), and N-(3-dimethylaminopropyl)-N-ethyl-carbodiimide (EDC) were all obtained from Aladdin Chemical Reagent Co. (Shanghai, China). Zirconyl chloride octahydrate (ZrOCl_2_·8H_2_O) was obtained from Macklin Biochemical Co. Ltd. (Shanghai, China). The SPE empty column (3 mL), polyethylene sieve plates (20 μm pore size), and commercial sorbents (C18, Carb, SAX, Florisil, NH_2_) were all supplied by Shenzhen Biocomma Biotech Co. (Shenzhen, China).

### 2.2. Apparatus

The SPE procedure was carried out on an SPE80 vacuum manifold equipped with 8-port model and a peristaltic pump (Jinan, China). The chromatographic separation and analysis of OPPs was performed on an Ultimate 3000 modular HPLC system with an automatic sampler (20 μL injection loop) and a UV-vis detector (Thermo Fisher Scientific, Waltham, MA, USA). A C18 column (Hypersil ODS2, 250 mm length × 4.6 mm i.d., 5 μm) was used for the separation of analytes. The mobile phase was composed of different proportions of methanol and water, the gradient elution was: 0–12 min, 73% methanol; 12–13 min, 73%–78% methanol. Two organic or water membranes (0.45 μm) were used to filter these mobile phases. The oven temperature was 25 °C. The flow rate of mobile phase was set at 1.0 mL min^−1^. The detection wavelength was 254 nm.

The synthesized materials were characterized using a series of instruments and equipment. A scanning electron microscope (SEM, JSM-6701F, JEOL, Tokyo, Japan) was used to observe the surface morphology. An IFS120HR Fourier transform infrared (FT-IR, Bruker, Karlsruhe, Germany) spectrometer (Thermo Fisher Scientific, Waltham, MA, USA) and an X-ray photoelectron spectrometer (XPS, ESCALAB 250Xi, Thermo Fisher Scientific, Waltham, MA, USA) were used to verify the component. A thermal gravimetric analyzer (TGA, STA449C, Netzsch, Selbu, Germany) was used to characterize the thermal stability. A BET analyzer (ASAP 2010, Micromeritics, Norcross, GA, USA) was used to obtain the surface area.

### 2.3. Preparation of Adsorption Materials

GO was firstly synthesized according to the modified Hummer’s method, which was descibed in the previous work [39]. Firstly, the SSM (Φ 8 mm) was immersed into the diluted hydrochloric acid to be etched, and after it was dried, it was put in APTES for 12 h to complete the modification of amino. Then, the obtained SSM was placed into a 5 mL centrifuge tube in addition to 4 mL of GO suspension (0.1 wt%), 0.001 g PDDA, 0.001 g EDC, and 0.001 g NHS, reacting for 2 h at 70 °C, which was then repeated 3 times to attain a coating with a desirable thickness. Furthermore, ZrO_2_ nanoparticles were deposited onto the surface of GO-grafted SSM through the reaction of ZrOCl_2_·8H_2_O in ammonia water, which was added into a reaction kettle for 12 h at 180 ℃. Finally, it was transferred to a freeze dryer (20 Pa, −50 °C), and after 10 h, the ZrO_2_/PDDA-GOA-SSM was obtained. Figure 1 illustrates a sequence of preparation steps of the preparation of ZrO_2_/PDDA-GOA-SSM.

The commercial sorbents including C18, -NH_2_, SAX, Carb, and Florisil, were removed from their original SPE column, and 50 mg of each of them were refilled into the new SPE column, respectively.

### 2.4. Sample Preparation and Extraction Procedure

Two vegetables (chives and pak choi) were bought from the local market (Qingdao, China). Firstly, they were twisted into powder in a high-speed pulverizer. One gram of the powder was put into an ethanol/water (10 mL, *v*/*v*, 70:30) and shaken for 3 h at 55 °C. After that, the mixture was centrifuged to remove the sample residue and retain the extractant. Then, the extractant was evaporated under the N_2_ stream. Finally, the residue was reconstituted with 20 mL of ultrapure water waiting for the extraction. In the study of relative recoveries, different volumes (2 μL, 4 μL, 10 μL) of the OPP standard solution (200 μg L^−1^) were added into the above 10 mL of mixture with ethanol/water and the sample.

The extraction procedure was performed on an SPE semi-automatic extraction apparatus. Three pieces of ZrO_2_/PDDA-GOA-SSM were filled into an empty plastic column, which was held with two sieve plates at each end of the cartridge. The prepared SPE column was preconditioned with 10 mL of water/acetonitrile (*v*/*v*, 1:1). A 20 mL aqueous sample containing three OPPs at the concentration of 200 μg L^−1^ was loaded onto the ZrO_2_/PDDA-GOA-SSM sorbent at a flow rate of 1.5 mL min^−1^. Then, 0.5 mL of water was used to wash the sorbent to remove other impurities. After drying for 5 min in air, the eluent (1.0 mL of acetonitrile) was used to elute the reserved OPPs on the ZrO_2_/PDDA-GOA-SSM at a flow rate of 0.8 mL min^−1^. Finally, the obtained extract was directly injected into the HPLC-UV system. Three replicate measurements were carried out to obtain the average value.

### 2.5. Adsorption Experiment

In the static-state adsorption experiment, one piece of ZrO_2_/PDDA-GOA-SSM was put into a 1.5 mL centrifuge tube with 1.0 mL of OPPs standards at different concentrations in the range of 5–30 μg mL^−1^. The adsorption time was 30 min and the process was carried out at room temperature. The adsorption capacity of ZrO_2_/PDDA-GOA-SSM towards OPPs was calculated and evaluated through the comparison of ZrO_2_/PDDA-GOA-SSM towards different OPPs. The calculation equation was as follows:
(1)$$q=\left(C_{i}-C_{f}\right)\times V_{s}$$
where *q* (μg) is the adsorption amount of OPP on one piece of ZrO_2_/PDDA-GOA-SSM; Vs is the volume of sample; and *C_i_* and *C_f_* (μg mL^−1^) are the initial and final concentrations of OPP, respectively.

The adsorption behavior of each OPP on the surface of ZrO_2_/PDDA-GOA-SSM was fitted by the Langmuir and Freundlich isotherm models, and their equations are as follows:
(2)$$\frac{C_{e}}{q_{e}}=\frac{C_{e}}{q_{max}}+\frac{1}{K_{L}q_{max}}$$
(3)$$q_{e}=K_{F}C_{e}^{1/n}$$
where *q_e_* (μg) and *q_max_* (μg) are the adsorption amount of OPP at the equilibrium and the maximum amount to form the single layer, respectively; *K_L_* (mL g μg^−1^) and *K_F_* [μg (mL μg^−1^)] are constants to reflect the adsorption capacity; *C_e_* (μg mL^−1^) is the concentration of OPP at the equilibrium; and n reflects the adsorption affinity of ZrO_2_/PDDA-GOA-SSM for OPPs.

A dimensionless constant (*R_L_*) called the separation factor or equilibrium parameter was defined as follows:$$R_{L}=\frac{1}{1+C_{0}K_{L}}$$
where *C_o_* is the highest initial concentration of OPPs. *R_L_* is a measure of the adsorption nature: *R_L_* = 0 (irreversible), 0 < *R_L_* < 1 (favorable), *R_L_* = 1 (linear), and *R_L_* > 1 (unfavorable).

## 3. Results

### 3.1. Characterization of the Sorbent

Figure 2a showed the photographs of the neet SSM and ZrO_2_/PDDA-GOA-modified SSMs. Compared with the bare SSM, it can be obviously observed that the GOA composite was successfully covered onto the surface of SSM. The surface morphology of ZrO_2_/PDDA-GOA-modified SSM was also investigated through the SEM characterization. Figure 2b showed the skeleton structure of SSM and the connection of stainless-steel wire to form the mesh structure. SEM images confirmed that the ZrO_2_ nanoparticle-modified graphene sheets were completely and uniformly coated onto the surface of every stainless-steel wire of SSM. Appendix A exhibits the SEM images of the etched SSM, PDDA-GOA-SSM, and ZrO_2_/PDDA-GOA-SSM. Compared to the etched SSM (Φ 50 μm), the surface of ZrO_2_/PDDA-GOA-SSM was rougher and more rugged, and the thickness of ZrO_2_/PDDA-GOA was approximately 5 μm. The BET surface area of the whole ZrO_2_/PDDA-GOA-coated SSM was characterized as 58.47 m^2^ g^−1^, which was higher than the surface area of 37.29 m^2^ g^−1^ of ZrO_2_/PDDA-GO-coated SSM.

The XPS analysis was performed to evaluate the successful synthesis of ZrO_2_/PDDA-GOA. As shown in Figure 3a, the C, O, and N elements all appeared in PDDA-GOA and ZrO_2_/PDDA-GOA. Besides, the peak of 183 eV (Zr3d) appeared in ZrO_2_/PDDA-GOA, which verified the successful modification of ZrO_2_.

The functional groups of ZrO_2_/PDDA-GOA were identified through the FT-IR spectroscopy. As can be seen in Figure 3b, the broad and strong peak observed at 3382 cm^−1^ was attributed to the O-H and N-H stretching vibrations from GOA and PDDA, respectively. The peaks at 2966 cm^−1^, 2929 cm^−1^, and 1463 cm^−1^ were referred to the stretching vibrations of saturated and unsaturated C-H bonds, and their bending vibration, respectively. The peak at 1612 cm^−1^ corresponded to the C=O stretching vibration of -COOH in GOA. The appearance of peak 1126 cm^−1^ was attributed to the stretching vibration of Zr-O-H groups. Characteristic peaks at 694 cm^−1^ and 873 cm^−1^ corresponded to the Zr-O bond as longitudinal and transverse modes, respectively. Meanwhile, the bending vibration of the benzene ring in GOA was in the range of 800–900 cm^−1^, so the peak at 873 cm^−1^ was relatively wider. The FT-IR spectrum further verified the successful preparation of the ZrO_2_/PDDA-GOA composite.

The thermal stability of ZrO_2_/PDDA-GOA was analyzed and assessed by thermogravimetric analysis (Figure 3c). The weight loss appeared below 100 °C and in the range of 100–300 °C, which was attributed to the loss of physisorbed and chemisorbed water, respectively. The weight loss ranging from 300 °C to 400 °C, and above 400 °C, was ascribed to the thermal decomposition of PDDA. The TGA curve indicated that the weight loss of the composite was mainly due to the water and a small amount of polymer, and the loading of ZrO_2_ on the surface of GOA was relatively stable.

### 3.2. Interaction Energy

The theoretical interaction energies between ZrO_2_ and three OPPs were calculated with the Gaussian 09 software based on the density functional theory. Through the repeat adjustment and optimization of acting sites and interaction models between ZrO_2_ and three OPPs, the interaction energies were calculated. In the calculated process, the π–π interaction, hydrogen bonding, and coordination bonding were all taken into consideration. As shown in Table 1, binding energies (ΔHF) of phoxim, temephos, and fenitrothion on the ZrO_2_ nanoparticle were -40.89, -34.03, and -46.93 kcal mol^−1^, respectively. The changes of Gibbs free energy (ΔG) were -26.04, -20.25, and -32.40 kcal mol^−1^, respectively. The coordination interaction between Zr and O was stronger than that of Zr and S. Therefore, compared with other two OPPs, fenitrothion possessing the P=O group showed higher binding energy with ZrO_2_ nanoparticles. The negative value of ΔG demonstrated that OPPs would all spontaneously combine with ZrO_2_ nanoparticles.

### 3.3. Optimization of the Extraction Procedure

In this experiment, three pieces of ZrO_2_/PDDA-GOA-SSM were selected to be packed into the SPE column. Using less pieces of mesh could bring about a relatively low adsorption amount, while using more pieces of mess could lead to the blocking of sieve plates and decrease the flow rate. Other experimental conditions including volume of the sample, sample pH, the rate of sample loading, the eluent, and volume of the eluent, were optimized to increase the extraction efficiency. In the optimization experiments, the deionized water was used to dilute the mother liquor of analytes to 200 μg L^−1^.

#### 3.3.1. Extraction Conditions

The volume of the sample should be optimized as a higher volume to improve the enrichment factor. However, due to the finite amount of extraction material and finite adsorption sites, the adsorption amount for analytes was limited. Different volumes of sample were loaded from 5 mL to 60 mL measured by the ZrO_2_/PDDA-GOA-SSM-SPE-HPLC-UV method. Figure 4a exhibits that the extraction recoveries remained almost unchanged until 20 mL of the sample, and after that, the extraction efficiency gradually decreased. Thus, 20 mL was considered as the optimum sample volume.

The effect of the sample loading rate on the extraction efficiency was two-sided. In general, a high rate of sample loading would decrease the operation time. However, due to the insufficient contact between analytes and the sorbent, analytes could run off at a high rate and a shorter time. On the contrary, a lower rate would cause the “back-extraction” appearance of analytes between the sorbent and the sample solution. As shown in Figure 4b, extraction recoveries of three OPPs first rose and then went down. Hence, 1.0 mL min^−1^ as the sample loading rate was selected.

Sample pH affected the existing state of analytes and the charge form of sorbents. A change of sample pH would bring about the gain or loss of protons. The investigated result is shown in Figure 4c. This indicates that OPPs in the molecular form would easily overflow from the sample matrix in the original solution. Hence, the pH value of the sample remained unchanged in the following experiment.

#### 3.3.2. Desorption Conditions

A suitable eluent was an effective parameter to increase the extraction efficiency. A desirable desorption solvent must sufficiently elute all analytes from the surface of extraction materials. Several commonly used solvents including acetonitrile, methanol, and acetone were investigated. Figure 4d shows that acetonitrile as the eluent exhibited the highest extraction efficiency among three types of eluent. Thus, acetonitrile was used as the eluent of OPPs desorbed from the ZrO_2_/PDDA-GOA-SSM.

In the desorption process, when the eluent flowed past the sorbent, a minimum of eluent volume should be used in order to save organic solvent under the complete desorption of analytes. As shown in Figure 4e, acetonitrile samples in the range of 0.5–1.5 mL were investigated, respectively. The result presented that 1.0 mL of acetonitrile would elute analytes sufficiently. Therefore, 1.0 mL of acetonitrile was used to eluent OPPs adsorbed in ZrO_2_/PDDA-GOA-SSM.

### 3.4. Extraction Performance and Extraction Mechanism

The ZrO_2_/PDDA-GO-coated SSM was also directly dispersed into the sample solution to extract OPPs, but it was found that several modified materials would fall down from the surface of SSM with the prolonging of operation time. Thus, the SPE method was used to prevent the problem, in which ZrO_2_/PDDA-GO-coated SSM was packed into the SPE column with two sieve plates at the ends.

The extraction performance of commercial sorbents (C18, -NH_2_, SAX, Carb, and Florisil) and prepared materials (APTES-SSM, PDDA-GO-SSM, PDDA-GOA-SSM, and ZrO_2_/PDDA-GO-SSM) for the extraction of three OPPs are exhibited in Figure 5. Compared with single APTES-modified SSM, extraction recoveries of the prepared ZrO_2_/PDDA-GO- and ZrO_2_/PDDA-GOA-modified SSM were obviously improved. Through the freeze-drying of GO, the forming of GA brought the increase of extraction efficiency. Compared with various commercial sorbents, it was observed that the extraction efficiencies of ZrO_2_/PDDA-GOA-SSM for the three OPPs were obviously increased, which indicated the superiority of the prepared composite.

To further evaluate the adsorption capacity of ZrO_2_/PDDA-GOA-SSM, the static-state binding experiment was investigated at different concentrations of OPPs. Figure 4f shows the adsorption capacity-concentration profiles of analytes. When the initial concentration of OPPs increased, the adsorption amount of each OPP also increased, which presented a saturation adsorption at the initial concentration of 15 μg mL^−1^, and then reached a plateau from 15 μg mL^−1^ to 30 μg mL^−1^. The relationship between the adsorption amount and initial concentration was fitted through two most common theoretical adsorption models (Langmuir and Freundlich isotherms). Equations (2) and (3) were used to calculate the adsorption parameters, which are listed in Appendix A. They exhibited higher correlation coefficients than Freundlich model, which indicates that the adsorption process of three OPPs onto the surface of ZrO_2_/PDDA-GOA-SSM was closer to the Langmuir model. Furthermore, it demonstrated that the target OPPs were adsorbed onto the surface of ZrO_2_/PDDA-GOA-SSM in the monolayer coverage. In the Langmuir model, RL values, as the equilibrium parameter, were 0.3144, 0.5082, and 0.3257 for phoxim, temephos, and fenitrothion, respectively. This indicates that the adsorption was favorable.

Relative factors affecting the extraction recovery are all exhibited in Figure 6. The adsorption capacity of sorbent for analytes and the hydrophilic-hydrophobic property of analytes (octanol-water partitioning coefficient, Log*K_ow_*) co-determine the extraction recovery. Variation tendencies of theoretical adsorption energy and the adsorption amount were almost consistent, and the adsorption capacity was phoxim > temephos > fenitrothion. This indicates that the extraction recovery was mainly affected by the hydrophilic-hydrophobic property of analytes.

### 3.5. Evaluation of the Method

Under the optimized experimental conditions, the analytical performance of the established ZrO_2_/PDDA-GOA-SSM-SPE-HPLC-UV method was evaluated, which included the linear range, enrichment factor (EF), limit of quantification (LOQ), limit of detection (LOD), repeatability, and reproducibility. As shown in Table 2, a good linearity of response was observed in the range of 1.0–200 μg L^−1^ for phoxim and fenitrothion, and in the range of 2.5–200 μg L^−1^ for temephos, with a correlation coefficient of 0.9952–0.9990. LODs were found to be as low as 0.2–1.0 μg L^−1^ for the three OPPs. EFs of the ZrO_2_/PDDA-GOA-SSM for the three OPPs were evaluated, which were 20.0, 20.9, and 18.7, respectively. Five consecutive extractions using a single-extraction column were performed in the standard solution, and relative standard deviations (RSDs) were in the range of 1.3%–7.6%. The reproducibility was estimated through the recoveries of five different prepared SSMs for OPPs, and RSDs were in the range of 6.4%–10.1%. Thus, the established ZrO_2_/PDDA-GOA-SSM possessed high-extraction recoveries as well as good repeatability.

### 3.6. Stability and Lifetime

The stability and lifetime of the sorbent are also important parameters for the practical application. The mechanical structure of the single GOA was unstable. When added with the PDDA, the GOA was not easy to drop from the surface of SSM. Appendix A shows the change of extraction recoveries obtained by the proposed ZrO_2_/PDDA-GOA-SSM after using it several times. After eight times, extraction recoveries decreased by about 5%–10%. This presented that the ZrO_2_/PDDA-GOA-SSM was relative stable, which means it could meet the requirement. Therefore, in this experiment, a ZrO_2_/PDDA-GOA-SSM-packed extraction column could be used eight times and then discarded.

### 3.7. Application in Real Sample

In order to verify the applicability of the ZrO_2_/PDDA-GOA-SSM-SPE-HPLC-UV method, two vegetables (chives and pak choi) were analyzed. Generally, with the spraying of pesticides, leafy vegetables possess more residual pesticides, so pak choi and chives as two leafy vegetables that are commonly consumed were selected as the target samples to be detected and analyzed. Figure 7 shows typical chromatographs of the two vegetables and samples spiked with different concentrations of standard solutions. There was no analyte detected in the real samples. The applicability of this method was also evaluated by the matrix effect and relative recoveries of analytes. Three-level concentrations of the standard solution (20 μg L^−1^, 40 μg L^−1^, 100 μg L^−1^) were added into the samples to calculate the relative recovery. As shown in Table 3, the relative recoveries were in the range of 51.3%–92.7%. The matrix effect was calculated by the equation: ME (%) = B/A × 100, where A was the slope of calibration curve in the standard solution and B was the slope of calibration curve of OPPs in the real samples. The ME values of phoxim, temephos, and fenitrothion were 46.5%, 90.1%, and 82.1% for the pak choi, and 75.2%, 68.6%, and 72.4% for the chives, respectively. The low ME value of 46.5% was due to the low concentration of OPPs. Other ME values exhibited the matrix weakening effect in the sample analysis. The results also show that, compared with other environmental samples, the matrix effect in food samples was relatively obvious due to the complex interferences.

The established ZrO_2_/PDDA-GOA-SSM-SPE-HPLC-UV method was compared with other reported methods with regard to the determination of OPPs. As listed in Appendix A, most of the reported works mainly focused on liquid samples, and there were relatively less studies concerning vegetables as solid samples. Compared to the proposed method, this method demonstrated medium linear range and LODs. However, the LOD of this method (0.2 μg L^−1^, 0.002 mg kg^−1^) was lower than the maximum residue limit (0.2 mg kg^−1^), which shows that it could meet the requirement for the limited detection of OPPs.

## 4. Conclusions

In this research, a ZrO_2_/PDDA-GOA-modified SSM was obtained through the chemical modification and freeze-drying, which was used for the extraction and determination of OPPs in vegetables. SSM, as a novel substrate, relatively reduced the accumulation of graphene. Compared with commercial extraction materials and other prepared materials (APTES-SSM, ZrO_2_/PDDA-GO-SSM), the ZrO_2_/PDDA-GOA-modified SSM exhibited higher extraction recoveries for analytes. The excellent extraction efficiency of this sorbent was attributed to the unique properties of GOA, the selective adsorption of ZrO_2_ for OPPs, and the characteristic permeability of the SSM substrate. This indicates that the cheap and accessible SSM, as an important component of extraction devices, could be used in the following applications. The established ZrO_2_/PDDA-GOA-SSM method could be successfully applied for the detection of OPPs in real samples, and this novel sorbent has been proved to be a promising candidate for the extraction of OPPs.

## Figures and Tables

**Figure 1 foods-10-01616-f001:**
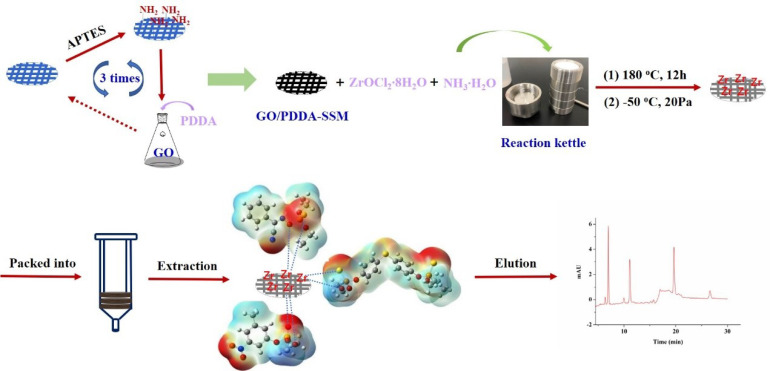
Preparation of the ZrO_2_/PDDA-GOA-modified SSM and the response signal.

**Figure 2 foods-10-01616-f002:**
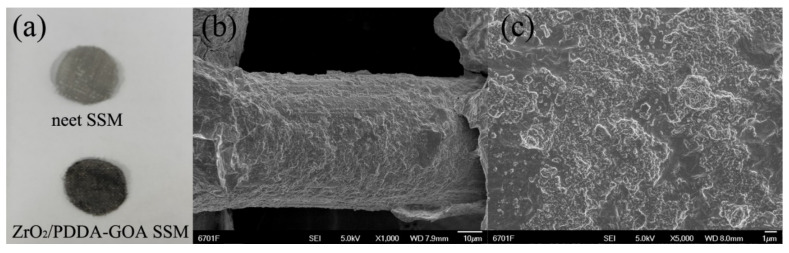
Photographs of the bare SSM and ZrO_2_/PDDA-GOA-modified SSM (**a**) and SEM images of ZrO2/PDDA-GOA-modified SSM with different magnifications (**b**,**c**).

**Figure 3 foods-10-01616-f003:**
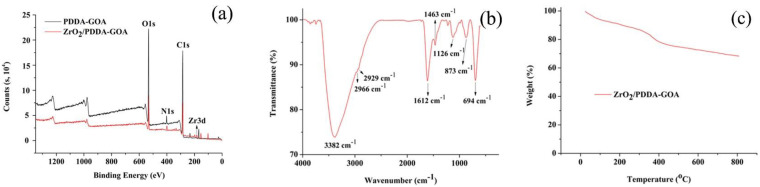
Characterization of extraction materials: XPS analysis (**a**), FT-IR spectrum (**b**), and TGA analysis (**c**).

**Figure 4 foods-10-01616-f004:**
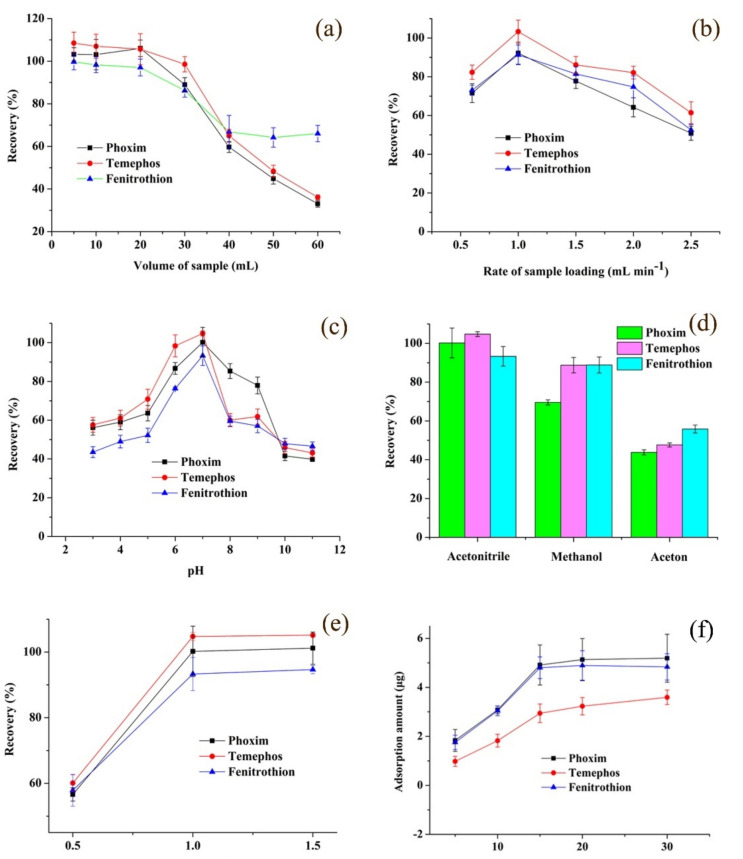
Effect of extraction and desorption conditions on the extraction performance: volume of sample (**a**), rate of sample loading (**b**), sample pH (**c**), type of eluent (**d**), volume of eluent (**e**); as well as the adsorption capacity-concentration profiles of analytes (**f**).

**Figure 5 foods-10-01616-f005:**
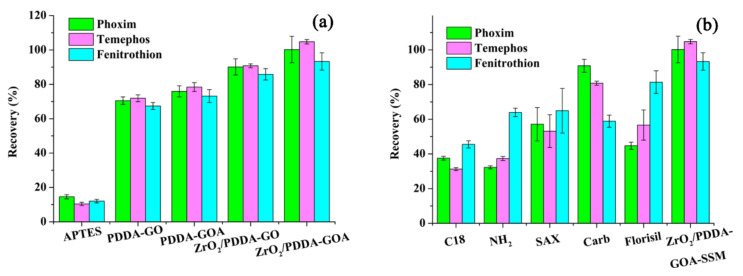
Comparison of extraction performance of different sorbents for analytes: (**a**) different prepared sorbent; (**b**) commercial sorbents.

**Figure 6 foods-10-01616-f006:**
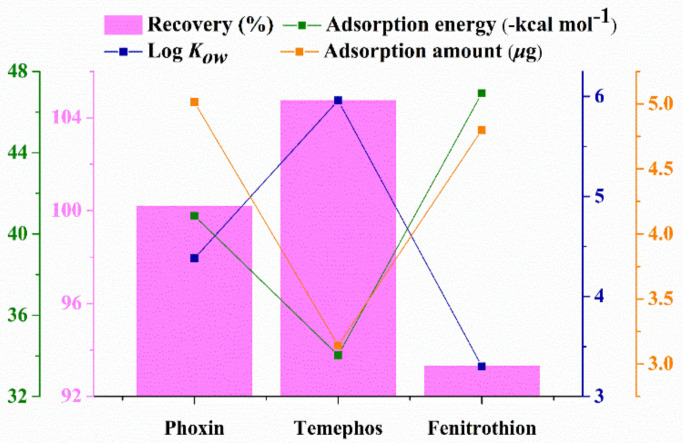
The variation tendency of different affecting factor.

**Figure 7 foods-10-01616-f007:**
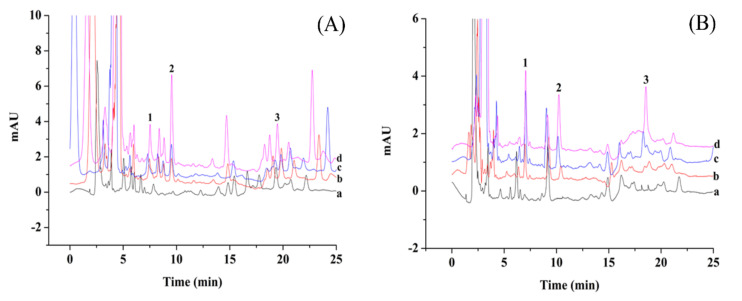
Chromatograms of pakvchoi (**A**) and chives (**B**), and samples (line **a**) spiked with standard solutions of 20 μg L^−1^ (line **b**), 40 μg L^−1^ (line **c**), and 100 μg L^−1^ (line **d**) by the proposed method.

**Table 1 foods-10-01616-t001:** Interaction energies of three OPPs on the ZrO_2_ nanoparticle and octanol-water partition coefficient (log *Kow*) of OPPs.

Analyte	Structure	ΔHF (kcal mol^−1^)	ΔG (kcal mol^−1^)	Log *K_ow_*
Phoxin	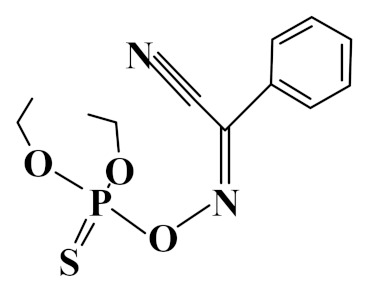	−40.9	−26.0	4.4
Temephos	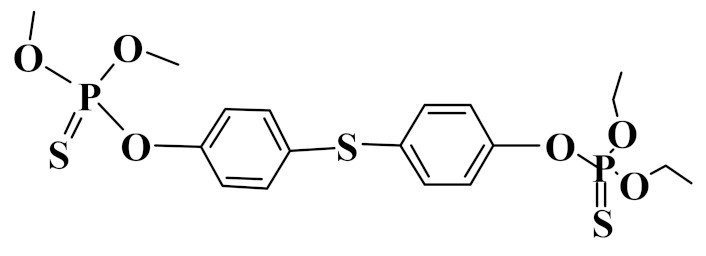	−34.0	−20.3	5.9
Fenitrothion	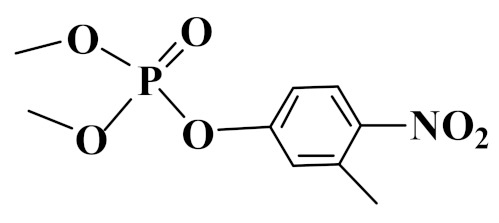	−46.9	−32.4	3.3

**Table 2 foods-10-01616-t002:** Analytical parameters for OPPs measured with the proposed ZrO_2_/PDDA-GOA-SSM-SPE-HPLC-UV method.

Compound	Linear Equation	Linear Range (μg L^−1^)	R ^a^	EF ^b^	LOQ ^c^ (μg L^−1^)	LOD ^d^ (μg L^−1^)	Repeatability (RSD ^e^, *n* = 5, %)	Reproducibility (RSD, *n* = 5, %)
Phoxim	y = 0.2329x + 0.2541	1–200	0.9952	20.0	1	0.5	7.6	10.1
Temephos	y = 0.2571x + 0.3962	2.5–200	0.9964	20.9	2.5	1.0	1.3	8.5
Fenitrothion	y = 0.1983x + 0.1864	1–200	0.9990	18.7	1	0.2	5.0	6.4

^a^ Correlation coefficient. ^b^ EF, enrichment factor. ^c^ LOQ, limit of quantification. ^d^ LOD, limit of detection for S/N = 3. ^e^ RSD, relative standard deviation.

**Table 3 foods-10-01616-t003:** Relative recoveries added with different concentrations of the standard solution in two vegetables.

Compound	Added (µg g^−1^)	Pak Choi		Chives	
		Recovery (%)	RSD (*n* = 3, %)	Recovery (%)	RSD (*n* = 3, %)
Phoxim	20	51.3	1.2	67.0	4.4
	40	69.8	5.3	92.7	8.3
	100	52.5	0.6	80.6	15.0
Temephos	20	91.9	0.6	71.7	6.0
	40	83.5	2.9	86.2	2.9
	100	91.8	5.3	63.8	2.0
Fenitrothion	20	80.3	5.5	53.6	3.4
	40	81.9	4.9	66.9	2.7
	100	78.9	1.8	73.4	2.2

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
