# Peer review of "ZrO2 Nanoparticles and Poly(diallyldimethylammonium chloride)-Doped Graphene Oxide Aerogel-Coated Stainless-Steel Mesh for the Effective Adsorption of Organophosphorus Pesticides"

_foods, 2021, doi:10.3390/foods10071616_

Round 1

Reviewer 1 Report

Dear Editor,

This manuscript describes the preparation of a stainless steel meshes coated with an aerogel composed of graphene oxide, a polymer and ZrO2 nanoparticles, which were then used to fabricate a SPE column by packing 3 pieces. The coated meshes were characterized by different techniques to demonstrate the uniform coating and the presence of all the components. The SPE devices were then used for the extraction of 3 organophosphate pesticides with different characteristics. The SPE procedure was optimized using a factor-by-factor approach, it was validated using standards prepared in water. The performance of the extraction phase was compared with other commercial phases and the possible extraction mechanisms and interactions between the analytes and the extraction material were assessed by computational and adsorption studies. Finally, the SPE cartridges were used to evaluate the matrix effects using 2 different vegetable sasmples.

The manuscript is well-organized and it is easy to follow. The introduction gives sufficient background on the topic to clarify the goal of the study, and the English is good. I think the study is interesting: the coating of a mesh is a great alternative to maximize the surface area of an extraction phase, which will contribute to enhance the extraction rate and capacity of any sorbent. Moreover, authors have included some interesting studies to elucidate the interactions between the analytes and the sorbent, which will help in the design of more selective extraction phases and will allow modulating the extraction paramters to obtain better performance.

However, I have an important concern regarding the use of the proposed coated-mehses. Authors discuss the development of stainless steel mesh-based sorptive extraction (reference 20 of the manuscript) as the basis for their study. In that work by Amiri and Ghaemi, the coated mesh is used in a extraction strategy similar to thin film microextraction and dispersive solid-phase extraction. Thus, taking advantage of the ridigity of the mesh, the high surface area of the coating and the magnetic susceptibility of the mesh, it was directly dispersed (using magnetic stirring) in the sample to carry out the extraction. However, in the present study, 3 pieces of the coated meshes were packed in a SPE cartridge. Why were they packed? Does this packing block the active sites of the meshes when they are pressed against each other? Would not be easier, simpler and more sustainable to directly disperse the mesh in the sample? And why 3 pieces? Was the number of pieces optimized?

Moreover, I found some other issues that I will list below:

  • Line 199, about SEM. It is clear that an uniform coating was obtained. However, it would interesting to include images of the mesh during the different steps of the coating procedure: neat mesh, coated with GO and PDDA, coated with GO, PDDA and ZrO2, and with and without the freeze drying step. This would help in confirming the correct coating of the mesh as well as the different morphology or porosity of the coating when the aerogel is obtained. Is it possible to estimate the thickness of the coating by comparing SEM images of the neat mesh and the coated mesh?

  • How were the measurements of the surface area accomplished? It would be interesting to obtain the values for the GO-based and the GOA-based meshes to compare and evidence the effect of the freeze drying process.

  • Line 259, regarding SPE optimization. Why does the absolute recovery decreases as the sample volume increase? Do authors have any clue about this behavior? Which desorption conditions were used when optimizing the extraction parameters?

  • Figure 4d, there is a mistake in the labels, the third batch of columns should be acetone instead of fenitrothion.

  • Section 3.4. In my opinion, when proposing composites as extraction phase, it is mandatory to demonstrate each component is required to obtain the desired analytical performance. Thus, in Figure 5 I miss the comparison with the mesh coated with PDDA-GO and PDDA-GOA without ZrO2 nanoparticles to demonstrate they are really affecting the extraction performance. Moreover, the difference between the mesh after the synthetic procedure and the aerogel is not significant. I think it would be also interesting compare the results with those obtained by directly packing the sorbent (without the mesh) to demonstrate the superiority of this configuration.

  • With respect to commercial SPE columns, neither in the expeimrental or the R&D section the characteristics of these cartridges are included. I think this is important for the comparison study in Figure 5. I guess the same conditions were used for all the SPE columns.

  • Table 2. I suggest including the calibration slopes in the table since it gives information about the sensitivity of the method. Limits of quantification too.

  • Line 343: please, correct SSW. According to the text, I guess the reproducibility was made with 5 different SPE columns, each of them prepared with 3 different meshes, right?

  • In Table S1, please, specify when it is µg/L or µg/kg. I suggest including enrichment factor values. I suggest including more studies using the same detection technique (HPLC-UV-Vis) since it is difficult to compare the results when GC and MS are used. Please, include the definition of the abbreviations for the methods. I would also include in the table if the sorbent or device could be reused. It would be also interesting to include if the methods require a pre-treatment of the samples (as it occurs in the present study) or the food sample is directly analyzed with the extraction method. I think authors must improve the discussion regarding the advantages of the proposed method, particularly considering QuEChERS has widely demonstrated to be a potential, reliable, simple and efficient method for the determination of pesticides from food samples.

  • Figure 7, please check the labelling for the different chromatograms with the figure caption. Lines 375-380: Why do authors think matrix effects are observed? Why these samples were selected as target samples?

  • Authors talk about selectivity in different sections of the manuscript (including abstract and conclusions). However, I do not think authors have demonstrated the selectivity of the coating considering: (i) the coated-mesh was only evaluated for the 3 target OPPs, (ii) the chromatograms obtained for the analysis of samples, which are really complex and demonstrate the extraction of numerous compounds from the sample. I think more studies should be carried out to include those “strong” statements.

Considering all these issues, I must recommend major revision of the manuscript since some important aspects of the manuscript must be addressed.

Author Response

Response to Reviewer 1:

This manuscript describes the preparation of a stainless steel meshes coated with an aerogel composed of graphene oxide, a polymer and ZrO2 nanoparticles, which were then used to fabricate a SPE column by packing 3 pieces. The coated meshes were characterized by different techniques to demonstrate the uniform coating and the presence of all the components. The SPE devices were then used for the extraction of 3 organophosphate pesticides with different characteristics. The SPE procedure was optimized using a factor-by-factor approach, it was validated using standards prepared in water. The performance of the extraction phase was compared with other commercial phases and the possible extraction mechanisms and interactions between the analytes and the extraction material were assessed by computational and adsorption studies. Finally, the SPE cartridges were used to evaluate the matrix effects using 2 different vegetable samples.

The manuscript is well-organized and it is easy to follow. The introduction gives sufficient background on the topic to clarify the goal of the study, and the English is good. I think the study is interesting: the coating of a mesh is a great alternative to maximize the surface area of an extraction phase, which will contribute to enhance the extraction rate and capacity of any sorbent. Moreover, authors have included some interesting studies to elucidate the interactions between the analytes and the sorbent, which will help in the design of more selective extraction phases and will allow modulating the extraction paramters to obtain better performance.

Response:First of all, we would like to express our sincere gratitude for your careful reading of our manuscript and pointing out the inadequacies, for your positive comments and helpful suggestions on our manuscript. Careful revisions have been made according to your suggestions in the revised manuscript, and point-by-point responses to each comment are also given. We hope that it would be more acceptable for publication to your remarkable insight. Thank you very much!

However, I have an important concern regarding the use of the proposed coated-mehses. Authors discuss the development of stainless steel mesh-based sorptive extraction (reference 20 of the manuscript) as the basis for their study. In that work by Amiri and Ghaemi, the coated mesh is used in a extraction strategy similar to thin film microextraction and dispersive solid-phase extraction. Thus, taking advantage of the ridigity of the mesh, the high surface area of the coating and the magnetic susceptibility of the mesh, it was directly dispersed (using magnetic stirring) in the sample to carry out the extraction. However, in the present study, 3 pieces of the coated meshes were packed in a SPE cartridge. Why were they packed? Does this packing block the active sites of the meshes when they are pressed against each other? Would not be easier, simpler and more sustainable to directly disperse the mesh in the sample? And why 3 pieces? Was the number of pieces optimized?

Response:Thank you very much for your careful reading and comment. One piece of the coated mesh possessed the finite adsorption amount towards analytes. We had tried to disperse the coated mesh in the sample solution, but found that some modified material would fall down from the stainless-steel mesh with the extraction time increased. Although the dropped material was not much, the extraction capacity would decrease gradually and further affect the lifetime of the coated meshes. The sieve plate in SPE cartridge would prevent the loss of extraction materials. Less pieces of meshes used would bring relatively low adsorption amount, while more pieces of meshes would be easy to lead to the blocking of sieve plates and decrease the flow rate. Therefore, in this experiment, three pieces of meshed were selected. Under the fixed sorbent, the volume of sample (the amount of analytes) were optimized.

Moreover, I found some other issues that I will list below:

  • Line 199, about SEM. It is clear that an uniform coating was obtained. However, it would interesting to include images of the mesh during the different steps of the coating procedure: neat mesh, coated with GO and PDDA, coated with GO, PDDA and ZrO2, and with and without the freeze drying step. This would help in confirming the correct coating of the mesh as well as the different morphology or porosity of the coating when the aerogel is obtained. Is it possible to estimate the thickness of the coating by comparing SEM images of the neat mesh and the coated mesh?

Response:Thank you very much for your comment. The below Figs.1-3 showed that the SEM images of the etched SSM, PDDA-GOA-coated SSM, and ZrO2/PDDA-GOA-coated SSM, respectively. In the preparation process, the composite of PDDA-GOA was obtained by the mixing of PDDA and GO, and the composite was assembled onto the surface of SSM, which was not assembled alone. Compared to the etched SSM (Φ 50 μm), the surface of ZrO2/PDDA-GOA-SSM was rougher and more rugged, and the thickness of ZrO2/PDDA-GOA was approximately 5 μm.

Fig. S1 SEM images of the etched SSM, PDDA-GOA-coated SSM, and ZrO2/PDDA-GOA-coated SSM.

  • How were the measurements of the surface area accomplished? It would be interesting to obtain the values for the GO-based and the GOA-based meshes to compare and evidence the effect of the freeze drying process.

Response:Thank you very much for your comment. The BET surface area of these extraction materials was measured by BET analyzer (ASAP 2010, Micromeritics, USA). The surface of ZrO2/PDDA-GO-based stainless-steel wire was 37.29 m2 g-1, which was lower than 58.47 m2 g-1 of ZrO2/PDDA-GOA-based stainless-steel wire. It presented that the effect of the freeze-drying process. Their relatively low surface area was resulted from the solid stainless-steel wire.

  • Line 259, regarding SPE optimization. Why does the absolute recovery decreases as the sample volume increase? Do authors have any clue about this behavior? Which desorption conditions were used when optimizing the extraction parameters?

Response:Thank you very much for your comment. The demonstrated had been presented in Paragraph 1 in Section 3.3.1. Due to the finite amount and finite adsorption sites of 3 pieces of stainless-steel wire, the adsorption amount of them towards OPPs was limited. When the adsorption amount reached the maximum, with the increase of sample volume, the extraction recovery would decrease. The figures of optimized conditions presented in manuscript, which was performed in optimized conditions, which was 20 mL of sample volume, 1.0 mL min-1 of sample loading rate, the original sample pH, 1.0 mL of acetonitrile as the eluent, the rate of elution of 0.5 mL min-1.

  • Figure 4d, there is a mistake in the labels, the third batch of columns should be acetone instead of fenitrothion.

Response:Thank you very much for your careful reading and comment. We were very regret for this mistake, and the corrected figure 4d was displayed in revised manuscript.

  • Section 3.4. In my opinion, when proposing composites as extraction phase, it is mandatory to demonstrate each component is required to obtain the desired analytical performance. Thus, in Figure 5 I miss the comparison with the mesh coated with PDDA-GO and PDDA-GOA without ZrO2 nanoparticles to demonstrate they are really affecting the extraction performance. Moreover, the difference between the mesh after the synthetic procedure and the aerogel is not significant. I think it would be also interesting compare the results with those obtained by directly packing the sorbent (without the mesh) to demonstrate the superiority of this configuration.

Response:Thank you very much for your comment and suggestion. Figure 5 had been revised, which included the APTES-SSM, PDDA-GO-SSM, PDDA-GOA-SSM, ZrO2/PDDA-GO-SSM, ZrO2/PDDA-GOA-SSM. Because the sorbent used the stainless-steel wire as the substrate, the extraction column without the stainless-steel wire did not exhibit any extraction recovery towards analytes. Compared the neet stainless-steel wire (APTES-SSM), the stainless-steel wire coated different sorbents exhibited different extraction recovery towards analytes. When GO was freeze-dried to from the GOA, the extraction recovery would increase. Through the investigation and comparison, the modification of ZrO2 also enhanced the extraction recoveries.

  • With respect to commercial SPE columns, neither in the expeimrental or the R&D section the characteristics of these cartridges are included. I think this is important for the comparison study in Figure 5. I guess the same conditions were used for all the SPE columns.

Response:Thank you very much for your comment. Experiment conditions of commercial sorbents for the extraction of OPPs were consistent with the prepared ZrO2/PDDA-GOA-SSM. These commercial sorbents were powder, and they were also 50 mg as the sorbent to be extracted and investigated.

  • Table 2. I suggest including the calibration slopes in the table since it gives information about the sensitivity of the method. Limits of quantification

Response:Thank you very much for your comment. The linear equation and limit of quantification were all listed in Table 2 as the following:

Compound

Linear

equation

Linear range (μg L-1)

Ra

EFb

LOQc

(μg L-1)

LODd

(μg L-1)

Repeatability (RSDe, n=5, %)

Reproducibility (RSD, n=5, %)

Phoxim

y=0.2329x+0.2541

1-200

0.9952

20.0

1

0.5

7.6

10.1

Temephos

y=0.2571x+0.3962

2.5-200

0.9964

20.9

2.5

1.0

1.3

8.5

Fenitrothion

y=0.1983x+0.1864

1-200

0.9990

18.7

1

0.2

5.0

6.4

Table 2. Analytical parameters for OPPs measured with the proposed ZrO2/PDDA-GOA-SSM based on SPE-HPLC-UV method.

aCorrelation coefficient. bEF, enrichment factor. cLOQ, limit of quantification. dLOD, limit of detection for S/N = 3. eRSD, relative standard deviation.

  • Line 343: please, correct SSW. According to the text, I guess the reproducibility was made with 5 different SPE columns, each of them prepared with 3 different meshes, right?

Response:Thank you very much for your comment. In line 343, the SSW had been replaced by “SSM”. As you said, the reproducibility had been investigated using 5 different SPE column with 3 pieces of meshes.

  • In Table S1, please, specify when it is µg/L or µg/kg. I suggest including enrichment factor values. I suggest including more studies using the same detection technique (HPLC-UV-Vis) since it is difficult to compare the results when GC and MS are used. Please, include the definition of the abbreviations for the methods. I would also include in the table if the sorbent or device could be reused. It would be also interesting to include if the methods require a pre-treatment of the samples (as it occurs in the present study) or the food sample is directly analyzed with the extraction method. I think authors must improve the discussion regarding the advantages of the proposed method, particularly considering QuEChERS has widely demonstrated to be a potential, reliable, simple and efficient method for the determination of pesticides from food samples.

Response:Thank you very much for your comment. According to your mentioned comments, Table S1 had been modified. And we had added and compared more reports about the HPLC-UV-Vis detection technique. The definition of the abbreviations for the methods had been listed below the Table. The lifetimes and enrichment factors had been also compared. The revised manuscript was as following:

Table S1 Comparison of methods used for the determination of OPPs.

Method

Sorbent/extractant

Sample

Sorbent

EF

Linearity

(μg L-1, μg kg-1)

LOD

(μg L-1, μg kg-1)

Lifetime (times)

Ref.

MSPEa-HPLC-UV

Poly(ionic liquid)/Fe3O4

Tea drink

60 mg

84-161

1-200 μg L-1

0.01 μg L-1

20

[1]

HF-SPMEb-HPLC-UV

MIL-101@GO

Tomato, cucumber and agricultural water

--

41-49

1-500 μg L-1

0.21-0.27 μg L-1

--

[2]

SPE-HPLC-UV

MIPc

Water

40 mg

330

50-1000 μg L-1

0.07-0.12 μg L-1

>50

[3]

MSPE-HPLC-UV

NiFe2O4@SiO2@polyaniline-ionic liquid

Fruit juice

15 mg

--

0.21-500 μg L-1

0.06-0.17 μg L-1

8

[4]

QuEChERSd-GC/MS

Multi-walled carbon nanotubes

Peanut oil

100 mg

--

5-200 μg kg-1

0.7-1.6 μg kg-1

--

[5]

MSDEe-GC/MS

C18

Bovine liver samples

500 mg

--

500-1500 μg kg-1

25-100 μg kg-1

--

[6]

HS-SPME-GC/MS

Polydimethylsiloxane/divinylbenzene

Milk, cows

--

--

14.6-32 μg L-1

2.16-10.85 μg L-1

--

[7]

DSPEf-GC/FID

Zinc-based metal organic framework

Water, fruit juice

8 mg

801-914

0.1-100 μg L-1

0.03-0.21 μg L-1

8

[8]

SPE-HPLC/UV

ZrO2/PDDA-GOA/SSM

Vegetables

--

18.7-20.9 

1-200 μg L-1

0.2 μg L-1

8

This study

aMSPE, magnetic solid-phase extraction; bHF-SPME, hollow fiber solid-phase microextraction; cMIP, molecularly imprinted polymer; dQuEChERS, Quick, Easy, Cheap, Effective, Rugged, Safe. eMSDE, matrix solid-phase dispersion; fDSPE, dispersive solid-phase extraction.

  • Figure 7, please check the labelling for the different chromatograms with the figure caption. Lines 375-380: Why do authors think matrix effects are observed? Why these samples were selected as target samples?

Response:Thank you very much for your comment. The figure7 caption had been replaced by “Figure 7. Chromatograms of pakchoi (A) and chives (B), and samples (a) spiked with standard solutions of 20 μg L-1 (b), 40 μg L-1 (c), and 100 μg L-1 (d) by the proposed method.”

Generally, with the spraying of pesticides, leafy vegetables possessed more residual pesticides, so pakchoi and chives as two leafy vegetables commonly consumed were selected as the target samples to be detected and analyzed. The food samples possessed many components, which often affected the detection of analytes, so the matrix effect was investigated through the standard addition method. In order to evaluate the matrix effect and the applicability of this method, relative recoveries of analytes were investigated and evaluated. Three-level concentrations of the standard solution (20 μg L-1, 40 μg L-1, 100 μg L-1) were added into the samples to calculate the relative recovery and the matrix effect.

  • Authors talk about selectivity in different sections of the manuscript (including abstract and conclusions). However, I do not think authors have demonstrated the selectivity of the coating considering: (i) the coated-mesh was only evaluated for the 3 target OPPs, (ii) the chromatograms obtained for the analysis of samples, which are really complex and demonstrate the extraction of numerous compounds from the sample. I think more studies should be carried out to include those “strong” statements.

Response:Thank you very much for your comment. According to your comment, we realized that the high selectivity had been overemphasized. Because many OPPs are banned by our country, their standard solution can not be sold. And we also considered the separation of the selected OPPs. So, three OPPs (phoxim, temephos, fenitrothion) had been regarded as the researched targets. In this experiment, on the basis of a strong affinity between ZrO2 and the phosphoric group, the sorbent had been designed for the extraction of OPPs. The theoretical interaction energies between ZrO2 and three OPPs were calculated. The coordination interaction between Zr and O was stronger than that of Zr and S. The prepared material exhibited definite adsorption selectivity towards the phosphoric compounds, but they were lack of adsorption selectivity for a organophosphorus compound. The desirable recoveries spiked with different standard solutions and matrix effects had been investigated and used to assess the selectivity. But according to your comment, we realized that the high selectivity had been overemphasized. Therefore, in the corresponding section, we had made changes.

Considering all these issues, I must recommend major revision of the manuscript since some important aspects of the manuscript must be addressed.

Response:We would like to express our heartfelt thanks for your positive comments and helpful suggestions on our manuscript again. Careful revisions have been made according to your suggestions in the revised manuscript, and point-by-point responses to each comment are also given. Some necessary editorial work has been done in the revised manuscript (additions are indicated in red and deletions in blue). We hope that it would be more acceptable to your remarkable insight. Thank you very much!

Reviewer 2 Report

Manuscript Number: foods-1246949

Title: ZrO2 Nanoparticles and Poly(diallyldimethylammonium chloride)-Doped Graphene Oxide Aerogel-coated Stainless-steel Mesh for the Effective Adsorption of Organophosphorus Pesticides

This article aimed to investigate the novel sorbent based on the ZrO2 nanoparticles and polydiallyldimethylammonium chloride modified graphene oxide aerogel grafted stainless steel mesh.  The novel sorbent was used for the extraction and determination of organophosphorus pesticides. This established method was successfully applied to detect organophosphorus pesticides in two vegetables (chives and pak choi).

I believe that this manuscript - article is a good contribution to research in this area.  

The objectives of this article were carried and are significant for further research.

Similar nanosorbents have been used to absorb e.g. uranium or heavy metals.

Menghui Zhao, Alemtsehay Tesfay Reda, and Dongxiang Zhang. Reduced Graphene Oxide/ZIF-67 Aerogel Composite Material for Uranium Adsorption in Aqueous Solutions ACS Omega 2020 5 (14), 8012-8022. DOI: 10.1021/acsomega.0c00089

Bhawna Sharma, Sourbh Thakur, Djalal Trache, Hamed Yazdani Nezhad, Vijay Kumar Thakur. Microwave-Assisted Rapid Synthesis of Reduced Graphene Oxide-Based Gum Tragacanth Hydrogel Nanocomposite for Heavy Metal Ions Adsorption. Nanomaterials 2020, 10 (8), 1616. https://doi.org/10.3390/nano10081616

Comments:

The subject of the manuscript is consistent with the scope of the journal Foods. There are no errors of fact or logic. The abstract does bring out the main points of the paper.

Information on the novel sorbent for the extraction and determination of organophosphorus pesticides is being processed clearly and concisely.

The literature references are adequate and recent.

Error in title: … Poly(diallyldimethylammonium chlride)…  correct to chloride

Figure 7 needs to be better explained!

The abbreviations MSPE, QuEChERS, MSDE, SPME, DSPE are not explained in the text of the manuscript, even if they are probably known to every analyst in the laboratory.

I don´t have other objections. 

So, the manuscript is need to minor revision.

Author Response

Response to Reviewer 2:

This article aimed to investigate the novel sorbent based on the ZrO2 nanoparticles and polydiallyldimethylammonium chloride modified graphene oxide aerogel grafted stainless steel mesh.  The novel sorbent was used for the extraction and determination of organophosphorus pesticides. This established method was successfully applied to detect organophosphorus pesticides in two vegetables (chives and pak choi).

I believe that this manuscript - article is a good contribution to research in this area.

The objectives of this article were carried and are significant for further research.

Similar nanosorbents have been used to absorb e.g. uranium or heavy metals.

Menghui Zhao, Alemtsehay Tesfay Reda, and Dongxiang Zhang. Reduced Graphene Oxide/ZIF-67 Aerogel Composite Material for Uranium Adsorption in Aqueous Solutions ACS Omega 2020 5 (14), 8012-8022. DOI: 10.1021/acsomega.0c00089

Bhawna Sharma, Sourbh Thakur, Djalal Trache, Hamed Yazdani Nezhad, Vijay Kumar Thakur. Microwave-Assisted Rapid Synthesis of Reduced Graphene Oxide-Based Gum Tragacanth Hydrogel Nanocomposite for Heavy Metal Ions Adsorption. Nanomaterials 2020, 10 (8), 1616. https://doi.org/10.3390/nano10081616

Response:Firstly, we would like to express our heartfelt thanks for your careful reading of our manuscript, and for your positive comments and helpful suggestions on our manuscript. Careful revisions have been made according to your suggestions in the revised manuscript, and point-by-point responses to each comment are also given. We have carefully revised the manuscript, and have tried our best to refine the language according to your suggestions and comments. Some necessary editorial work has been done in revised manuscript (additions are indicated in red and deletions in blue). We hope that it would be more acceptable to your remarkable insight. Thank you very much!

  These two related references had been added in Ref [26,27].

Comments:

The subject of the manuscript is consistent with the scope of the journal Foods. There are no errors of fact or logic. The abstract does bring out the main points of the paper.

Information on the novel sorbent for the extraction and determination of organophosphorus pesticides is being processed clearly and concisely.

The literature references are adequate and recent.

Error in title: … Poly(diallyldimethylammonium chlride)… correct tochloride

Response:Thank you very much for your comment. We are very sorry for this error. In revised manuscript, “Poly(diallyldimethylammonium chlride” had been replaced by “Poly(diallyldimethylammonium chloride”.

Figure 7 needs to be better explained!

Response:Thank you very much for your comment. The figure7 caption had been replaced by “Figure 7. Chromatograms of pakchoi (A) and chives (B), and samples (a) spiked with standard solutions of 20 μg L-1 (b), 40 μg L-1 (c), and 100 μg L-1 (d) by the proposed method.”

The abbreviations MSPE, QuEChERS, MSDE, SPME, DSPE are not explained in the text of the manuscript, even if they are probably known to every analyst in the laboratory.

Response:Thank you very much for your comment. The abbreviations MSPE, QuEChERS, MSDE, SPME, DSPE had been deleted in revised manuscript, and their full names had been added in Table S2 below.

I don´t have other objections. 

So, the manuscript is need to minor revision.

Response:We would like to express our heartfelt thanks for your positive comments and helpful suggestions on our manuscript again. We hope that it would be more acceptable to your remarkable insight. Thank you very much!

Round 2

Reviewer 1 Report

Dear Editor,

Authors have improved the manuscript and have addressed most of my comments but only in the responses. As a reviewer, when asking questions about the study, I expect some of the responses are also included in the manuscript. Asking about some aspects of the study it is because they are missing in the main manuscript and, in my opinion, they should be included. So, I think some information should be added in the text, such as why these samples were selected, why SPE was used as extraction technique, etc.

I have some comments for this new version:

  • I think it should be stated somewhere in the abstract that the extraction procedure is a solid-phase extraction method, using a cartridge packed with the meshes. The same for the last paragraph of the introduction.
  • Moreover, authors have included in the responses some information justifying why the meshes were packed instead of dispersed. I think this information should be included in the R&D section, somewhere at the end of the 3.1 section (maybe). I think it is important to clarify why this extraction technique was selected and why 3 pieces were used. As it is presented, it seems it was randomnly selected.
  • Please, include in the manuscript the BET surface area value for the coated mesh without the freeze-drying step. This is important because it justifies the presence of this step in the preparation procedure.
  • In the responses “Because the sorbent used the stainless-steel wire as the substrate, the extraction column without the stainless-steel wire did not exhibit any extraction recovery towards analytes”. Does it mean that the main responsible for the extraction is the SS wire? This statement is confusing.
  • Please, include somewhere in the text how the columns with commercial sorbents were prepared. At least, specify the amount that was packed.
  • Again, the discussion comparing the characteristics of the method with those already reported in the literature (Table S2) is weak. Despite new information in the table has been added, the discssion is the same. I still do not see the advatanes of the proposed method, particularly considering the parameters of the methods from referenes 3, 4 and 8 of Table S2 are better.

Author Response

Authors have improved the manuscript and have addressed most of my comments but only in the responses. As a reviewer, when asking questions about the study, I expect some of the responses are also included in the manuscript. Asking about some aspects of the study it is because they are missing in the main manuscript and, in my opinion, they should be included. So, I think some information should be added in the text, such as why these samples were selected, why SPE was used as extraction technique, etc.

Response:First of all, we would like to express our sincere gratitude for your careful reading of our manuscript and pointing out the inadequacies, for your positive comments and helpful suggestions on our manuscript. Careful revisions have been made according to your suggestions in the revised manuscript, and point-by-point responses to each comment are also given. We hope that it would be more acceptable for publication to your remarkable insight. Thank you very much!

The selected reason of these samples had been added in Section 3.7, which was “Generally, with the spraying of pesticides, leafy vegetables possessed more residual pesticides, so pakchoi and chives as two leafy vegetables commonly consumed were selected as the target samples to be detected and analyzed.”

The selected reason of SPE method had been presented in Section 3.4, which was “The ZrO2/PDDA-GO coated SSM was also directly dispersed into the sample solution to extract OPPs, but it found that several modified materials would fall down from the surface of SSM with the prolonging of operation time. So, the SPE method was used to prevent the problem, in which ZrO2/PDDA-GO coated SSM was packed into the SPE column with two sieve plates at ends.”

I have some comments for this new version:

  • I think it should be stated somewhere in the abstract that the extraction procedure is a solid-phase extraction method, using a cartridge packed with the meshes. The same for the last paragraph of the introduction.

Response:Thank you very much for your comment. The related information of using SPE method had added in Abstract and Introduction.

Abstract: “The ZrO2/PDDA-GOA-SSM was packed into the solid-phase extraction (SPE) cartridge to extract OPPs.” “It indicated that the ZrO2/PDDA-GOA-SSM-SPE-HPLC method could effectively extract and detect OPPs in vegetables.”

Introduction: “A SPE column was packed with ZrO2/PDDA-GOA-SSM to extract OPPs.” “The applicability of these established ZrO2/PDDA-GOA-SSM-SPE-HPLC method was also evaluated.”

  • Moreover, authors have included in the responses some information justifying why the meshes were packed instead of dispersed. I think this information should be included in the R&D section, somewhere at the end of the 3.1 section (maybe). I think it is important to clarify why this extraction technique was selected and why 3 pieces were used. As it is presented, it seems it was randomnly selected.

Response:Thank you very much for your comment and suggestion. The related content had been added in Section 3.3 and Section 3.4.

Section 3.3: “In this experiment, three pieces of ZrO2/PDDA-GOA-SSM were selected to be packed into the SPE column. Because less pieces of meshes used would bring relatively low adsorption amount, while more pieces of meshes would be easy to lead to the blocking of sieve plates and decrease the flow rate.”

Section 3.4: “The ZrO2/PDDA-GO coated SSM was also directly dispersed into the sample solution to extract OPPs, but it found that several modified materials would fall down from the surface of SSM with the prolonging of operation time. So, the SPE method was used to prevent the problem, in which ZrO2/PDDA-GO coated SSM was packed into the SPE column with two sieve plates at ends.”

  • Please, include in the manuscript the BET surface area value for the coated mesh without the freeze-drying step. This is important because it justifies the presence of this step in the preparation procedure.

Response:Thank you very much for your comment and suggestion. The BET surface area value for the coated mesh without the freeze-drying step had been added in Section 3.1, which was “The BET surface area of the whole ZrO2/PDDA-GOA coated SSM was characterized as 58.47 m2 g-1, which was higher than the surface area of 37.29 m2 g-1 of ZrO2/PDDA-GO coated SSM.”

  • In the responses “Because the sorbent used the stainless-steel wire as the substrate, the extraction column without the stainless-steel wire did not exhibit any extraction recovery towards analytes”. Does it mean that the main responsible for the extraction is the SS wire? This statement is confusing.

Response:Thank you very much for your comment. We are very sorry for the confusing demonstration. The unmodified or bare stainless-steel mesh had no extraction capacity towards OPPs due to the smooth surface. In the experiment, in order to obtain the rough surface, HCl had been used to etch the stainless-steel mesh.

  • Please, include somewhere in the text how the columns with commercial sorbents were prepared. At least, specify the amount that was packed.

Response:Thank you very much for your comment. The related content had been added in Section 2.3, which was “The commercial sorbents including C18, -NH2, SAX, Carb, Florisil, were removed from their original SPE column, and 50 mg of them were refilled into the new SPE column, respectively.”

  • Again, the discussion comparing the characteristics of the method with those already reported in the literature (Table S2) is weak. Despite new information in the table has been added, the discssion is the same. I still do not see the advatanes of the proposed method, particularly considering the parameters of the methods from referenes 3, 4 and 8 of Table S2 are better.

Response:Thank you very much for your comment. According to the related information of the previous reports and our work, an objective explanation was given again, which was removed to Section 3.7. The revised content was “The established ZrO2/PDDA-GOA-SSM-SPE-HPLC-UV method was compared with other reported methods with regard to the determination of OPPs. As listed in Table S2, most of the reported works mainly focused on the liquid sample, and the study concerning vegetables as the solid samples was relatively less. Compared to the proposed method, this method demonstrated medium linear range and LODs. But the LOD of this method (0.2 μg L-1, 0.002 mg kg-1) was lower than the maximum residue limit (0.2 mg kg-1), which presented that it could meet the requirement for the limited detection of OPPs.”
